# Compositional Image Decomposition with Diffusion Models

## Abstract

Given an image of a natural scene, we are able to quickly decompose it into a set of components such as objects, lighting, shadows, and foreground. We can then picture how the image would look if we were to recombine certain components with those from other images, for instance producing a scene with a set of objects from our bedroom and animals from a zoo under the lighting conditions of a forest, even if we have never seen such a scene in real life before. We present a method to decompose an image into such compositional components. Our approach, Decomp Diffusion, is an unsupervised method which, when given a single image, infers a set of different components in the image, each represented by a diffusion model. We demonstrate how components can capture different factors of the scene, ranging from global scene descriptors (*e.g.*, shadows, foreground, facial expression) to local scene descriptors (*e.g.*, objects). We further illustrate how inferred factors can be flexibly composed, even with factors inferred from other models, to generate a variety of scenes sharply different than those seen in training time.

## 1 Introduction

Humans have the remarkable ability to quickly learn new concepts, such as learning to use a new tool after observing just a few demonstrations (Allen et al., 2020). This skill relies on the ability to combine and reuse previously acquired concepts to accomplish a given task (Lake et al., 2017). This is particularly evident in natural language, where a limited set of words can be infinitely combined under grammatical rules to express various ideas and opinions (Chomsky, 1965). In this work, we propose a method to discover compositional concepts from images in an unsupervised manner, which may be flexibly combined both within and across different image modalities.

Prior works on unsupervised compositional concept discovery may be divided into two separate categories. One line of approach focuses on discovering a set of global, holistic factors by representing data points in fixed factorized vector space (Vedantam et al., 2018; Higgins et al., 2018; Singh et al., 2019; Peebles et al., 2020). Individual factors, such as facial expression or hair color, are represented as independent dimensions of the vector space, with recombination between concepts corresponding to recombination between underlying dimensions. However, since the vector space has a fixed dimensionality, multiple instances of a single factor, such as multiple different sources of lighting, may not be easily combined. Furthermore, as the vector space has a fixed underlying structure, individual factored vector spaces from different models trained on different datasets may not be combined, *e.g.*, the lighting direction in one dataset with the foreground of an image from another.

An alternative approach decomposes a scene into a set of different underlying "object" factors. Each individual factor represents a separate set of pixels in an image defined by a disjoint segmentation mask (Burgess et al., 2019; Locatello et al., 2020b; Monnier et al., 2021; Engelcke et al., 2021a). Composition between different factors then corresponds to composing their respective segmentation masks. However, this method struggles to model higher-level relationships between factors, as well as multiple global factors that collectively affect the same image.

Recently, COMET (Du et al., 2021a) proposes to instead decompose a scene into a set of factors represented as *energy functions*. Composition between factors corresponds to solving for a minimal energy image subject to each energy function. Each individual energy function can represent global concepts such as facial expression or hair color as well as local concepts such as objects. However, COMET is unstable to train due to second-order gradients, and often generates blurry images.

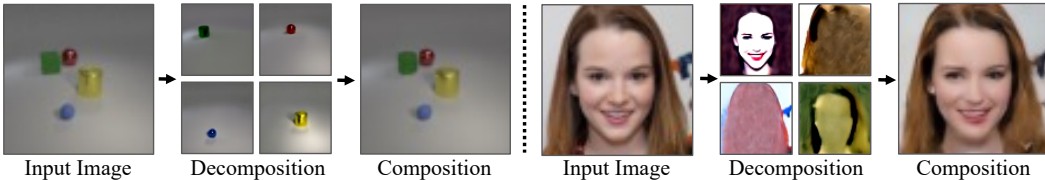

| Input Image | Decomposition | Composition | Input Image | Decomposition | Composition |

Figure 1: **Image Decomposition with Decomp Diffusion.** Our unsupervised method can decompose an input image into both local factors, such as objects (**Left**), and global factors (**Right**), such as facial features. Additionally, our approach can combine the deduced factors for image reconstruction.

In this paper, we leverage the close connection between Energy-Based Models (LeCun et al., 2006; Du & Mordatch, 2019) and diffusion models (Sohl-Dickstein et al., 2015; Ho et al., 2020) and propose Decomp Diffusion, an approach to decompose a scene into a set of factors, each represented as separate diffusion models. Composition between factors is achieved by sampling images from a composed diffusion distribution (Liu et al., 2022; Du et al., 2023), as illustrated in Figure 1. Similar to composition between energy functions, this composition operation allows individual factors to represent both global and local concepts and further enables the recombination of concepts across different models and datasets.

However, unlike the underlying energy decomposition objective of COMET, Decomp Diffusion may directly be trained through denoising, a stable and less expensive learning objective, and leads to higher resolution images. In summary, we contribute the following: First, we present Decomp Diffusion, an approach using diffusion models to decompose scenes into a set of different compositional concepts which substantially outperforms prior work using explicit energy functions. Second, we show Decomp Diffusion is able to successfully decompose scenes into both global concepts as well as local concepts. Finally, we show that concepts discovered by Decomp Diffusion generalize well, and are amenable to compositions across different modalities of data, as well as components discovered by other instances of Decomp Diffusion.

## 2 UNSUPERVISED DECOMPOSITION OF IMAGES INTO ENERGY FUNCTIONS

In this section, we introduce background information about COMET (Du et al., 2021a) which our approach extends. COMET infers a set of latent factors from an input image, and uses each inferred latent to define a separate energy function over images. To generate an image that exhibits inferred concepts, COMET runs an optimization process over images on the sum of different energy functions.

In particular, given an image $\boldsymbol{x} \in \mathbb{R}^D$, COMET uses a learned encoder $\text{Enc}_\phi(\boldsymbol{x})$ to infer a set of $K$ different latents $\boldsymbol{z}_k \in \mathbb{R}^M$, where each latent $\boldsymbol{z}_k$ represents a different concept in an image. Both images and latents are passed into an energy function $E_\theta(\boldsymbol{x}, \boldsymbol{z}_k) : \mathbb{R}^D \times \mathbb{R}^M \to \mathbb{R}$, which maps these variables to a scalar energy value.

Given a set of different factors $\boldsymbol{z}_k$, decoding these factors to an image corresponds to solving the optimization problem:

$$\underset{\boldsymbol{x}}{\text{argmin}} \sum_k E_\theta(\boldsymbol{x}; \boldsymbol{z}_k). \tag{1}$$

To solve this optimization problem, COMET runs an iterative gradient descent procedure from an image initialized from Gaussian noise. Factors inferred from either different images or even different models may likewise be decoded by optimizing the energy function corresponding to sum of energy function of each factor.

COMET is trained so that the $K$ different inferred factors $\boldsymbol{z}_k$ from an input image $\boldsymbol{x}_i$ define $K$ energy functions, so that the minimal energy state corresponds to the original image $\boldsymbol{x}_i$:

$$\mathcal{L}_{\text{MSE}}(\theta) = \left\| \underset{\boldsymbol{x}}{\text{argmin}} \left( \sum_k E_\theta(\boldsymbol{x}; \boldsymbol{z}_k) \right) - \boldsymbol{x}_i \right\|^2, \tag{2}$$

where $\boldsymbol{z}_k = \text{Enc}_\phi(\boldsymbol{x}_i)[k]$. The argmin of the sum of the energy functions is approximated by $N$ steps of gradient descent

$$\boldsymbol{x}_i^N = \boldsymbol{x}_i^{N-1} - \gamma \nabla_{\boldsymbol{x}} \sum_k E_\theta(\boldsymbol{x}_i^{N-1}; \text{Enc}_\phi(\boldsymbol{x}_i)[k]), \tag{3}$$

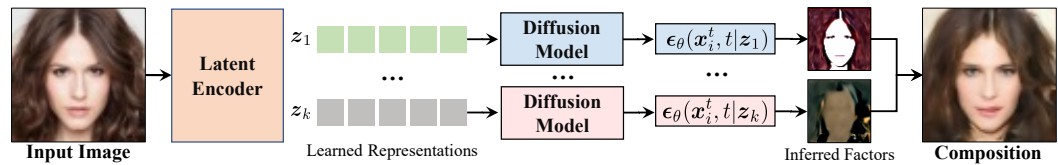

Figure 2: **Compositional Image Decomposition**. We learn to decompose each input image into a set of denoising functions $\{\epsilon_\theta(\boldsymbol{x}_i^t, t, |\boldsymbol{z}_k)\}$ representing $K$ factors, which can be composed to reconstruct the input.

where $\gamma$ is the step size. Optimizing the training objective in Equation 2 corresponds to back-propagating through this optimization objective. The resulting process is computationally expensive and unstable to train, as it requires computing second-order gradients.

## 3   COMPOSITIONAL IMAGE DECOMPOSITION WITH DIFFUSION MODELS

Next, we discuss how we may instead decompose images into a set of composable diffusion models. We first discuss how diffusion models may be seen as parameterizing energy functions in Section 3.1. Then in Section 3.2, we describe how we use this connection in Decomp Diffusion to decompose images into a set of composable diffusion models.

### 3.1   DENOISING NETWORKS AS ENERGY FUNCTIONS

Denoising Diffusion Probabilistic Models (DDPMs) (Sohl-Dickstein et al., 2015; Ho et al., 2020) are a class of generative models that facilitate generation of images $\boldsymbol{x}_0$ by iteratively denoising an image initialized from Gaussian noise. Given a randomly sampled noise value $\epsilon \sim \mathcal{N}(0, 1)$, as well as a set of $t$ different noise levels $\epsilon^t = \sqrt{\beta_t}\epsilon$ added to a clean image $\boldsymbol{x}_i$, a denoising model $\epsilon_\theta$ is trained to denoise the image at each noise level $t$:

$$\mathcal{L}_{\text{MSE}} = \|\epsilon - \epsilon_\theta(\sqrt{1 - \beta_t}\boldsymbol{x}_i + \sqrt{\beta_t}\epsilon, t)\|_2^2. \tag{4}$$

In particular, the denoising model learns to estimate a gradient field of natural images, describing the direction noisy images $\boldsymbol{x}^t$ with noise level $t$ should be refined to become natural images (Ho et al., 2020). As discussed in both (Liu et al., 2022; Du et al., 2023), this gradient field also corresponds to the gradient field of an energy function

$$\epsilon_\theta(\boldsymbol{x}^t, t) = \nabla_{\boldsymbol{x}} E_\theta(\boldsymbol{x}) \tag{5}$$

that represents the relative log-likelihood of a datapoint.

To generate an image from the diffusion model, a sample $\boldsymbol{x}^T$ at noise level $T$ is initialized from Gaussian noise $\mathcal{N}(0, 1)$ and then iteratively denoised through

$$\boldsymbol{x}^{t-1} = \boldsymbol{x}^t - \gamma\epsilon_\theta(\boldsymbol{x}^t, t) + \xi, \quad \xi \sim \mathcal{N}(0, \sigma_t^2 I), \tag{6}$$

where $\sigma_t^2$ is the variance according to a variance schedule and $\gamma$ is the step size[1]. This directly corresponds to the noisy energy optimization procedure

$$\boldsymbol{x}^{t-1} = \boldsymbol{x}^t - \gamma\nabla_{\boldsymbol{x}} E_\theta(\boldsymbol{x}^t) + \xi, \quad \xi \sim \mathcal{N}(0, \sigma_t^2 I). \tag{7}$$

The functional form of Equation 7 is very similar to Equation 3, and illustrates how sampling from a diffusion model is similar to optimizing a learned energy function $E_\theta(\boldsymbol{x})$ that parameterizes the relative negative log-likelihood of the data density.

When we train a diffusion model to recover a conditional data density that consists of a single image $\boldsymbol{x}_i$, *i.e.*, when we are autoencoding an image given an inferred intermediate latent $\boldsymbol{z}$, then the denoising network directly learns an $\epsilon_\theta(\boldsymbol{x}, \boldsymbol{t}, \boldsymbol{z})$ that estimates gradients of an energy function $\nabla_{\boldsymbol{x}} E_\theta(\boldsymbol{x}, \boldsymbol{z})$. This energy function has minimum

$$\boldsymbol{x}_i = \underset{\boldsymbol{x}}{\operatorname{argmin}} \, E_\theta(\boldsymbol{x}, \boldsymbol{z}), \tag{8}$$

as the highest log-likelihood datapoint will be $\boldsymbol{x}_i$. The above equivalence suggests that we may directly use diffusion models to parameterize the unsupervised decomposition of images into the energy functions discussed in Section 2.

---

[1] An linear decay $\frac{1}{\sqrt{1-\beta_t}}$ is often also applied to the output $\boldsymbol{x}^{t-1}$ for sampling stability.

## 3.2 Decompositional Diffusion Models

In COMET, given an input image $x_i$, we are interested in inferring a set of different latent energy functions $E_\theta(x, z_k)$ such that

$$x_i = \operatorname*{argmin}_x \sum_k E_\theta(x, z_k).$$

Using the equivalence between denoising networks and energy function discussed in Section 3.1 to recover the desired set of energy functions, we may simply learn a set of different denoising functions to recover an image $x_i$ using the objective

$$\mathcal{L}_{\mathrm{MSE}} = \left\| \epsilon - \sum_k \epsilon_\theta \left( \sqrt{1 - \beta_t} x_i + \sqrt{\beta_t} \epsilon, t, z_k \right) \right\|_2^2, \tag{9}$$

where each individual latent $z_k$ is inferred by a jointly learned neural network encoder $\mathrm{Enc}_\phi(x_i)[k]$. Since the encoder compresses the input $x_i$ into a set of low dimensional latent representations $z = \{z_1, z_2, \cdots, z_K\}$, by information bottleneck, each individual $z_k$ is encouraged to capture important, orthogonal information from the inputs, which we demonstrate correspond to factors such as objects or attributes of a scene. This resulting objective is simpler to train than that of COMET, as it requires only a single step denoising supervision and does not need computation of second-order gradients.

**Reconstruction Training.** As discussed in (Ho et al., 2020), the denoising network $\epsilon_\theta$ may either be trained to directly estimate the starting noise $\epsilon$ or the original image $x_i$. These two predictions are functionally identical, as $\epsilon$ can be directly obtained by taking a linear combination of $x^t$ and $x_i$. While standard diffusion training directly predicts $\epsilon$, we find that predicting $x_i$ and then regressing $\epsilon$ leads to better performance, as this training objective is more similar to autoencoder training.

Once we have recovered these denoising functions, we may directly use the noisy optimization objective in Equation 7 to sample from compositions of different factors. The full training and sampling algorithm for our approach are shown in Algorithm 1 and Algorithm 2 respectively.

---

**Algorithm 1** Training Algorithm

1: **Input:** Encoder $\mathrm{Enc}_\phi$, denoising model $\epsilon_\theta$, components $K$, data distribution $p_D$
2: **while** not converged **do**
3:     $x_i \sim p_D$
4:     ▷ *Extract components $z_k$ from $x_i$*
5:     $z_1, \ldots, z_K \leftarrow \mathrm{Enc}_\phi(x_i)$
6:     ▷ *Compute denoising direction*
7:     $\epsilon \sim \mathcal{N}(0, 1), t \sim \mathrm{Unif}(\{1, \ldots, T\})$
8:     $x_i^t = \sqrt{1 - \beta_t} x_i + \sqrt{\beta_t} \epsilon$
9:     $\epsilon_{\mathrm{pred}} \leftarrow \sum_k \epsilon_\theta(x_i^t, t, z_k)$
10:    ▷ *Optimize objective $\mathcal{L}_{MSE}$ wrt $\theta$:*
11:    $\Delta\theta \leftarrow \nabla_\theta \| \epsilon_{\mathrm{pred}} - \epsilon \|^2$
12: **end while**

---

**Algorithm 2** Image Generation Algorithm

1: **Input:** Diffusion steps $T$, denoising model $\epsilon_\theta$, latent vectors $\{z_1, \ldots, z_K\}$, step size $\gamma$
2: $x^T \sim \mathcal{N}(0, 1)$
3: **for** $t = T, \ldots, 1$ **do**
4:     ▷ *Sample Gaussian noise*
5:     $\xi \sim \mathcal{N}(0, 1)$
6:     ▷ *Compute denoising direction*
7:     $\epsilon_{\mathrm{pred}} \leftarrow \sum_k \epsilon_\theta(x^t, t, z_k)$
8:     ▷ *Run noisy gradient descent*
9:     $x^{t-1} = \frac{1}{\sqrt{1 - \beta_t}}(x^t - \gamma \epsilon_{\mathrm{pred}} + \sqrt{\beta_t} \xi)$
10: **end for**
11: **return** $x^0$

---

## 4 Experiments

In this section, we begin by comparing our approach for decomposing images with prior approaches. We evaluate how effectively each method decomposes individual components from images, representing both local and global factors in the scenes. As factors are discovered in an unsupervised manner, we name each factor based on visual inspection and provide an extensive set of examples to aid in visualization of each factor. We further assess the quality of image reconstruction as well as the underlying disentanglement of factors. Furthermore, we demonstrate the recombination of individual components to generate novel combinations. We analyze this across both within and across different image datasets.

### 4.1 Quantitative Metrics

For quantitative evaluation of image quality and disentanglement, we employ the following metrics:

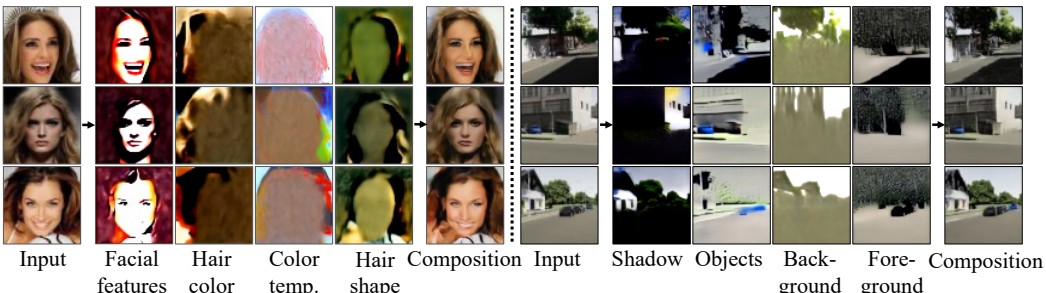

Figure 3: **Global Factor Decomposition.** Our method can enable global factor decomposition and reconstruction on CelebA-HQ (**Left**) and Virtual KITTI 2 (**Right**). Note that we name inferred concepts for easy understanding.

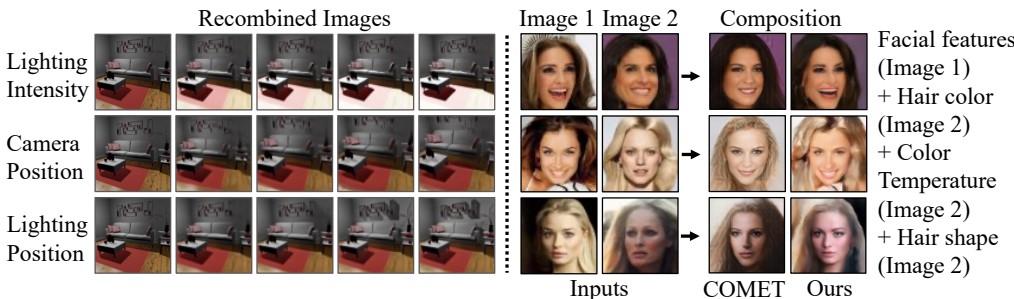

Figure 4: **Global Factor Recombination.** Recombination of inferred factors on Falcor3D and CelebA-HQ datasets. In Falcor3D (**Left**), we show image variations by varying inferred factors such as lighting intensity. In CelebA-HQ (**Right**), we recombine factors from two different inputs to generate novel face combinations.

**FID** (Heusel et al., 2017). Fréchet Inception Distance (FID) measures the quality of generative models based on the feature similarity between generated images and ground truth images, where image features are extracted using a pre-trained Inception model (Szegedy et al., 2016).

**KID** (Bińkowski et al., 2018). Kernel Inception Distance (KID) is an enhanced version of FID that performs well even when the number of generated samples is limited. While FID is sensitive to the number of generated samples, KID exhibits better behavior in such cases.

**LPIPS** (Zhang et al., 2018). LPIPS is a perceptual metric that measures the similarity of images by computing their distances in the feature space. A lower LPIPS score indicates a higher similarity.

**MIG** (Chen et al., 2018). The Mutual Information Gap (MIG) measures disentanglement quality using the mutual information between a latent variable and a ground truth factor.

**MCC** (Hyvärinen & Morioka, 2016). The Mean Correlation Coefficient (MCC) is another disentanglement quantitative evaluation. It matches each latent with a desired ground truth factor using the correlation matrix between the ground truth factors and latent representations.

To evaluate the quality of reconstructed images, we use FID, KID and LPIPS on images reconstructed from CelebA-HQ (Karras et al., 2017), Falcor3D (Nie et al., 2020). Virtual KITTI 2 (Cabon et al., 2020) and CLEVR (Johnson et al., 2017). Following COMET (Du et al., 2021a), we evaluate the learned latent representations on disentanglement using both MIG and MCC on the Falcor3D dataset.

## 4.2 GLOBAL FACTORS

Given a set of input images, we illustrate that our unsupervised approach can capture a set of global scene descriptors such lighting and background, and recombine them to construct image variations. We evaluate results in terms of image quality and disentanglement of global components.

**Decomposition and Reconstruction.** In Figure 3, we show how our approach decomposes CelebA-HQ face images into a set of factors on the left-hand side. These factors include facial features, hair color, skin tone, and hair shape, each named based on qualitative visualization. In addition, we further compare our method on image reconstruction with existing baselines in Figure 5. Our method generates better reconstructions than COMET, in that images are sharper and more similar to the input as well as other recent baselines.

| Model | CelebA-HQ | | | Falcor3D | | | Virtual KITTI 2 | | | CLEVR | | |
|---|---|---|---|---|---|---|---|---|---|---|---|---|
| | FID ↓ | KID ↓ | LPIPS ↓ | FID ↓ | KID ↓ | LPIPS ↓ | FID ↓ | KID ↓ | LPIPS ↓ | FID ↓ | KID ↓ | LPIPS ↓ |
| $\beta$-VAE ($\beta = 4$) | 107.29 | 0.107 | 0.239 | 116.96 | 0.124 | 0.075 | 196.68 | 0.181 | 0.479 | 316.64 | 0.383 | 0.651 |
| MONet | 35.27 | 0.030 | 0.098 | 69.49 | 0.067 | 0.082 | 67.92 | 0.043 | 0.154 | 60.74 | 0.063 | 0.118 |
| COMET | 62.64 | 0.056 | 0.134 | 46.37 | 0.040 | 0.032 | 124.57 | 0.091 | 0.342 | 103.84 | 0.119 | 0.141 |
| Slot Attention | 56.41 | 0.050 | 0.154 | 65.21 | 0.061 | 0.079 | 153.91 | 0.113 | 0.207 | 27.08 | 0.026 | 0.031 |
| Hessian Penalty | 34.90 | 0.021 | – | 322.45 | 0.479 | – | 116.91 | 0.084 | – | 25.40 | 0.016 | – |
| GENESIS-V2 | 41.64 | 0.035 | 0.132 | 130.56 | 0.130 | 0.097 | 134.31 | 0.105 | 0.202 | 318.46 | 0.403 | 0.631 |
| Ours | **16.48** | **0.013** | **0.089** | **14.18** | **0.008** | **0.028** | **21.59** | **0.008** | **0.058** | **11.49** | **0.011** | **0.012** |

Table 1: **Image Reconstruction Evaluation.** We evaluate the quality of $64 \times 64$ reconstructed images using FID, KID and LPIPS on $10,000$ images from $4$ different datasets. Our method achieves the best performance.

On the right side of Figure 3, we also demonstrate that Decomp Diffusion can be applied to infer factors such as shadow, lighting, landscape, and objects on Virtual KITTI 2. We can further compose these factors to reconstruct the input images, as illustrated in the rightmost column. Comparative decompositions from other methods can be found in Figure XVIII.

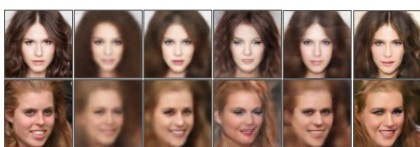

Input  $\beta$-VAE  SlotAttn  COMET  GEN-V2  Ours

Figure 5: **Reconstruction comparison.** Our method can reconstruct input images with a high fidelity on CelebA-HQ.

We further provide qualitative results to demonstrate the impact of number of concepts $K$ on both CelebA-HQ and Falcor3d in Figure XVI and Figure XVII. As expected, different $K$ can lead to different sets of decomposed concepts being produced, but certain concepts are learned across different $K$, such as the facial features concepts in Figure XVII.

**Recombination.** In Figure 4, we provide additional insights into each captured factor by recombining the decomposed factors from both the Falcor3D and CelebA-HQ datasets. On the left-hand side, we demonstrate how recombination can be performed on a source image by varying a target factor, such as lighting intensity, while preserving the other factors. This enables us to generate image variations using inferred factors such as lighting intensity, camera position, and lighting position.

| Dataset | Multiple Components | Predict $x_0$ | MSE ↓ | LPIPS ↓ | FID ↓ | KID ↓ |
|---|---|---|---|---|---|---|
| CelebA-HQ | Yes | No | 105.003 | 0.603 | 155.46 | 0.141 |
| | No | Yes | 88.551 | 0.192 | 30.10 | 0.022 |
| | Yes | Yes | **76.168** | **0.089** | **16.48** | **0.013** |
| CLEVR | Yes | No | 56.179 | 0.3061 | 42.72 | 0.033 |
| | No | Yes | 26.094 | 0.2236 | 24.27 | 0.023 |
| | Yes | Yes | **6.178** | **0.0122** | **11.54** | **0.010** |

Table 2: **Ablations.** We analyze the impact of predicting $x_0$ or $\epsilon$, as well as using multiple components or a single component. We compute pixel-wise MSE and LPIPS of reconstructions on both CLEVR and CelebA-HQ.

On the right-hand side of Figure 4, we further show how factors extracted from different human faces can be recombined to generate a novel human face that exhibits selected global factors. For instance, we can combine facial features from one person with hair shape from another to create a new face that exhibit chosen properties. These results illustrate that our method can effectively disentangle images into global factors that can be recombined for novel generalization.

**Quantitative results.** To quantitatively compare different methods, we first evaluate the disentanglement of the given methods on the Falcor3D dataset. As shown in Table 3, Decomp Diffusion (dim = 64) achieves the best scores across disentanglement metrics, showing its effectiveness in capturing a set of global scene descriptors. In addition, we evaluate our models with different latent dimensions of 32, 64, and 128 to verify the impact of latent dimension. We find that our method achieves the best performance when using a dimension of 64. We posit that a smaller dimension may lack the capacity to encode all the information, thus leading to worse disentanglement. A larger dimension may be too large that it fails to separate distinct factors. Thus, we apply PCA to project the output dimension 128 to 64 (last row), and we observe that it can boost the MIG performance but lower the MCC score.

Finally, we evaluate the visual quality of reconstructed images using the decomposed scene factors, as presented in Table 1. We observe that our method outperforms existing methods in terms of FID, KID and LPIPS across datasets, indicating superior image reconstruction quality.

**Diffusion Parameterizations.** We next analyze two choices of diffusion parameterizations, *e.g.*, whether the model should predict $x_0$ or the noise $\epsilon$, in Table 2. We find that directly predicting the input $x_0$ (3rd and 6th rows) outperforms the $\epsilon$ parametrization (1st and 4th row) on both CelebA-HQ and CLEVR datasets in terms of MSE and LPIPS (Zhang et al., 2018). This is due to using a reconstruction-based training procedure, as discussed in Section 3.2. We also compare using a single component to learn reconstruction (2nd and 5th rows) with our method (3rd and 6th rows), which uses

| Model | Dim ($D$) | $\beta$ | Decoder Dist. | MIG ↑ | MCC ↑ |
|---|---|---|---|---|---|
| InfoGAN | 64 | – | – | $2.48 \pm 1.11$ | $52.67 \pm 1.91$ |
| $\beta$-VAE | 64 | 4 | Bernoulli | $8.96 \pm 3.53$ | $61.57 \pm 4.09$ |
| $\beta$-VAE | 64 | 16 | Gaussian | $9.33 \pm 3.72$ | $57.28 \pm 2.37$ |
| $\beta$-VAE | 64 | 4 | Gaussian | $10.90 \pm 3.80$ | $66.08 \pm 2.00$ |
| GENESIS-V2* | 128 | – | – | $5.23 \pm 0.02$ | $63.83 \pm 0.22$ |
| MONet | 64 | – | – | $13.94 \pm 2.09$ | $65.72 \pm 0.89$ |
| COMET | 64 | – | – | $19.63 \pm 2.49$ | $76.55 \pm 1.35$ |
| Ours | 32 | – | – | $11.72 \pm 0.05$ | $57.67 \pm 0.09$ |
| Ours | 64 | – | – | $\mathbf{26.45 \pm 0.16}$ | $\mathbf{80.42 \pm 0.08}$ |
| Ours | 128 | – | – | $12.97 \pm 0.02$ | $80.27 \pm 0.17$ |
| Ours* | 128 | – | – | $16.57 \pm 0.02$ | $71.19 \pm 0.15$ |

Table 3: **Disentanglement Evaluation.** Mean and standard deviation (s.d.) metric scores across 3 random seeds on the Falcor3D dataset. Decomp Diffusion enables better disentanglement according to 2 common disentanglement metrics. The asterisk (*) indicates PCA is applied to project the output dimension to 64.

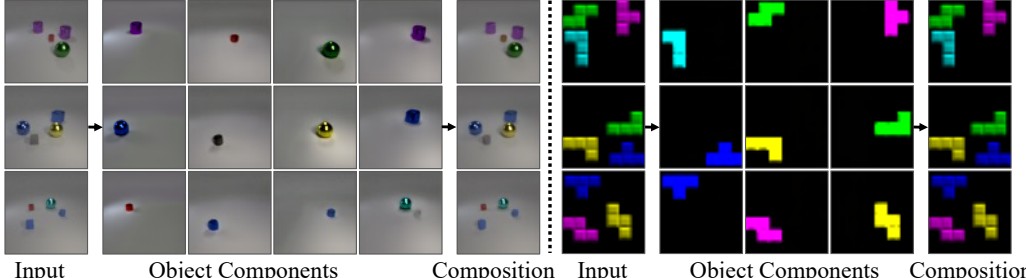

Input     Object Components     Composition     Input     Object Components     Composition

Figure 6: **Local Factor Decomposition.** Illustration of object-level decomposition on CLEVR (**left**) and Tetris (**right**). Our method can extract individual object components that can be reused for image reconstruction.

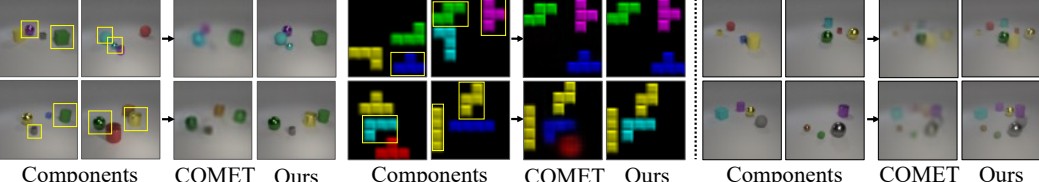

Components   COMET  Ours     Components   COMET  Ours     Components   COMET  Ours

Figure 7: **Local Factor Recombination.** We recombine local factors from 2 images to generate composition of inferred object factors. On both CLEVR and Tetris (**Left**), we recombine inferred object components in the bounding box to generate novel object compositions. On CLEVR (**Right**), we compose all inferred factors to generalize up to 8 objects, though training images only contain 4 objetcs.

multiple components for reconstruction. Our method demonstrates the best reconstruction quality, as measured by MSE and LPIPS.

## 4.3 LOCAL FACTORS

Given an input image with multiple objects, *e.g.*, a purple cylinder and a green cube, we aim to factorize the input image into individual object components using object-level segmentation.

**Decomposition and Reconstructions.** We qualitatively evaluate local factor decomposition on object datasets such as CLEVR and Tetris in Figure 6. Given an image with multiple objects, our method can isolate each individual object components, and can also faithfully reconstruct the input image using the set of decomposed object factors. Note that since our method does not obtain an explicit segmentation mask per object, it is difficult to quantitatively assess segmentations (but we found our approach to almost always correctly segment objects).

**Recombination.** To further validate our approach, we present qualitative results showcasing the recombination of captured local factors from different input images to generate previously unseen image combinations. In Figure 7, we demonstrate how our method utilizes a subset of factors from each image for local factor recombination. On the left-hand side, we show the generation of novel object combinations using factorized energy functions representing individual local object components from two inputs, shown within the bounding boxes, on both the CLEVR and Tetris datasets. On the right-hand side, we demonstrate how our method can recombine all existing local components from two CLEVR images, even though each training image only consists of 4 objects.

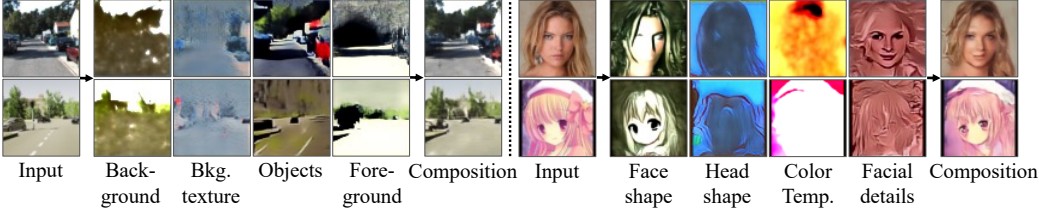

Input · Back-ground · Bkg. texture · Objects · Fore-ground · Composition · Input · Face shape · Head shape · Color Temp. · Facial details · Composition

Figure 8: **Multi-modal Dataset Decomposition.** We show our method can capture a set of global factors that are shared between hybrid datasets such as KITTI and Virtual KITTI 2 scenes (**Left**), and CelebA-HQ and Anime faces (**Right**). Note that we name inferred concepts for better understanding.

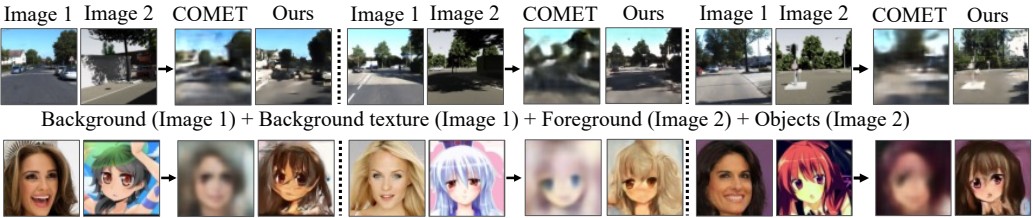

Image 1 · Image 2 · COMET · Ours · Image 1 · Image 2 · COMET · Ours · Image 1 · Image 2 · COMET · Ours

Background (Image 1) + Background texture (Image 1) + Foreground (Image 2) + Objects (Image 2)

Head shape (Image 1) + Color temperature (Image 1) + Face shape (Image 2)  + Facial details (Image 2)

Figure 9: **Multi-modal Dataset Recombination.** Our method exhibits the ability to recombine inferred factors from various hybrid datasets. We can recombine different extracted factors to generate unique compositions of KITTI and Virtual KITTI 2 scenes (**Top**), and compositions of CelebA-HQ and Anime faces (**Bottom**).

Thus, our method generalizes well to novel combinations of 8 object components. We illustrate that our approach is highly effective at recombining local factors to create novel image combinations.

## 4.4 CROSS DATASET GENERALIZATION

We next assess the ability of our approach to extract and combine concepts across multiple datasets. We investigate the recombination of factors in multi-modal datasets, and the combination of separate factors from distinct models trained on different datasets.

**Multi-modal Decomposition and Reconstruction.** We evaluate our proposed method's efficacy in decomposing multi-modal datasets into a set of factors. Because such datasets are comprised of images from different modalities, they pose a challenge for extracting a common set of factors. However, as shown in Figure 8, our method successfully performs this task. The left-hand side exhibits the decomposition of images from a hybrid dataset comprising KITTI and Virtual KITTI into a set of global factors, such as background, lighting, and shadows. The right-hand side decomposes the two types of faces into a cohesive set of global factors including face shape, hair shape, hair color, and facial details, which can be utilized for reconstruction. This demonstrates our method's effectiveness in factorizing hybrid datasets into a set of factors.

**Multi-modal Recombination.** Furthermore, we assess the ability of our proposed method to recombine obtained factors across multi-modal datasets, as illustrated in Figure 9. On the top half, from the hybrid KITTI and Virtual KITTI dataset, we recombine extracted factors from two distinct images to produce novel KITTI-like scenes, for instance incorporating a blue sky background with shadows in the foreground. In the bottom half, we present our method's capacity for reusing and combining concepts to generate unique anime faces. Specifically, we combine hair shapes and colors from a human face image with face shape and details from an anime face image, resulting in novel anime-like faces.

**Cross Dataset Recombination.** Given two instances of trained models, where one is trained on CLEVR objects and CLEVR Toy objects, we investigate how we can combine local factors extracted from different modalities to generate novel combinations. In Figure 10, our method can extract object components in the bounding box from two images from different datasets, and then further combine to generate unseen combinations of object components from different models. In Table V, we provide the FID and KID scores of generated recombinations against the original CLEVR dataset and CLEVR Toy dataset. Our method outperforms COMET on both datasets, indicating the model can obtain better visual quality and more cohesive recombination.

Figure 10: **Cross Dataset Recombination.** We further showcase our method's ability to recombine across datasets using 2 different models that train on CLEVR and CLEVR Toy, respectively. We compose inferred factors as shown in the bounding box from two different modalites to generate unseen compositions.

## 5 RELATED WORK

**Compositional Generation.** An increasing body of recent work has studied compositional generation (Du et al., 2020; Liu et al., 2021; 2022; Wu et al., 2022; Shi et al., 2023; Cong et al., 2023; Cho et al., 2023; Du et al., 2023; Huang et al., 2023; Nie et al., 2021; Wang et al., 2023; Gandikota et al., 2023), where we seek to generate outputs subject to a set of different conditions. Existing work on compositional generation focus either on modifying the underlying generative process to focus on a set of specifications (Feng et al., 2022; Shi et al., 2023; Cong et al., 2023; Huang et al., 2023), or by composing a set of independent models specifying desired constraints (Du et al., 2020; Liu et al., 2021; 2022; Nie et al., 2021; Du et al., 2023; Wang et al., 2023). Similar to (Du et al., 2021b), our work aims discover a set of compositional components from an unlabeled dataset of images which may further be integrated with compositional operations from (Du et al., 2023; Liu et al., 2022).

**Unsupervised Decomposition.** Our work is related to existing research on unsupervised decomposition, where works have studied how to obtain global factor disentanglement (Higgins et al., 2017; Burgess et al., 2018; Locatello et al., 2020a; Klindt et al., 2021; Peebles et al., 2020; Singh et al., 2019). These approaches typically focus on discovering a global latent space which best describes the input space, with prior work in (Preechakul et al., 2022) also exploring this global latent space on diffusion models. Our approach aims instead to decompose data into multiple different compositional vector spaces, which allow us to both compose multiple instances of one factor together, as well as compose factors across different datasets. The most similar work in this direction is COMET (Du et al., 2021a), but unlike COMET we decompose images into a set of different diffusion models, and illustrate how this enables higher fidelity and more scalable image decomposition.

Our work is also related to the field of unsupervised object discovery (Burgess et al., 2019; Greff et al., 2019; Locatello et al., 2020b; Lin et al., 2020; Engelcke et al., 2021a; Du et al., 2021b; Singh et al., 2022; Kipf et al., 2022; Seitzer et al., 2022; Jia et al., 2023), which seeks to decompose a scene into a set of different objects. Developed concurrently with our approach, Jiang *et al.* (Jiang et al., 2023) and Wu *et al.* (Wu et al., 2023) proposes to decomposes images into a set of object-centric diffusion models. Separate from these works, our approach does not assume the explicit decomposition of images into segmented components, enabling the ability to represent objects and global factors in a scene, drawing on the connection of diffusion models and EBMs.

## 6 CONCLUSION

**Limitations.** Our work has several limitations. First, our current approach decomposes images into a fixed number of factors that is specified by the user. While there are cases where the number of components is apparent, in many datasets the number is unclear and there may be variable number dependent on image. In Section C, the sensitivity of our approach to the number of components specified is studied and we find that we recover duplicate components when the number is too large, and subsets of components when it is too small. We believe a principled approach to determining number of factors is a interesting direction of future work. In addition, factors discovered by approach is not guaranteed be distinct from the original image or from each other, and if the latent encoder's embedding dimension is too large, each latent factor may capture the original image itself. Adding explicit regularization to enforce independence between latents would be interesting future work.

**Conclusion.** In this work, we present Decomp Diffusion and demonstrate its efficacy at decomposing images into both global factors of variation, such as facial expression, lighting, and background, and local factors, such as constituent objects. We further illustrate the ability of different inferred components to compose across multiple datasets and models. We hope that our work inspires future research in unsupervised discovery of compositional representations in images.

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

## A OVERVIEW

In this supplementary material, we present additional qualitative results for various domains in Section B. Next, we describe the model architecture for our approach in Section D. Finally, we include experiment details on training datasets, baselines, training, and inference in Section E.

## B ADDITIONAL RESULTS

We first provide additional results on global factor decomposition and recombination in Section B.1. We then give additional results on object-level decomposition and recombination in Section B.2. Finally, we provide more results that demonstrate cross-dataset generalization in Section B.3.

### B.1 GLOBAL FACTORS

**Decomposition and Reconstruction.** In Figure XI, we present supplemental image generations that demonstrate our approach's ability to capture global factors across different domains, such as human faces and scene environments. The left side of the figure displays how our method can decompose images into global factors like facial features, hair color, skin tone, and hair shape, which can be further composed to reconstruct the input images. On the right, we show additional decomposition and composition results using Virtual KITTI 2 images. Our method can effectively generate clear, meaningful global components from input images. In Figure XII, we show decomposition and composition results on Falcor3D data. Through unsupervised learning, our approach can accurately discover a set of global factors that include foreground, background, objects, and lighting.

**Recombination.** Figure XIII showcases our approach's ability to generate novel image variations through recombination of inferred concepts. The left-hand side displays results of the recombination process on Falcor3D data, with variations on lighting intensity, camera position, and lighting position. On the right-hand side, we demonstrate how facial features and skin tone from one image can be combined with hair color and hair shape from another image to generate novel human face image combinations. Our method demonstrates great potential for generating diverse and meaningful image variations through concept recombination.

### B.2 LOCAL FACTORS

**Decomposition and Reconstruction.** We present additional results for local scene decomposition in Figure XIV. Our proposed method successfully factorizes images into individual object components, as demonstrated in both CLEVR (**Left**) and Tetris (**Right**) object images. Our approach also enables the composition of all discovered object components for image reconstruction.

**Recombination.** We demonstrate the effectiveness of our approach for recombination of local scene descriptors extracted from multi-object images such as CLEVR and Tetris. As shown in Figure XV, our method is capable of generating novel combinations of object components by recombining the extracted components (shown within bounding boxes for easy visualization). Our approach can effectively generalize across images to produce unseen combinations.

### B.3 CROSS DATASET GENERALIZATION

We investigate the recombination of factors inferred from multi-modal datasets, and the combination of separate factors extracted from distinct models trained on different datasets.

**Multi-modal Decomposition and Reconstruction.** We further demonstrate our method's capability to infer a set of factors from multi-modal datasets, *i.e.*, a dataset that consists of different types of images. On the left side of Figure XXI, we provide additional results on a multi-modal dataset that consists of KITTI and Virtual KITTI 2. On the right side, we show more results on a multi-modal dataset that combines both CelebA-HQ and Anime datasets.

**Multi-modal Recombination.** In Figure XXII, we provide additional recombination results on the two multi-modal datasets of KITTI and Virtual KITTI 2 on the left hand side of the Figure, and CelebA-HQ and Anime datasets on the right hand side of the Figure.

**Cross Dataset Recombination.** We also show more results for factor recombination across two different models trained on different datasets. In Figure XXIII, we combine inferred object components from a model trained CLEVR images and components from a model trained on CLEVR Toy images. Our method enables novel recombinations of inferred components from two different models.

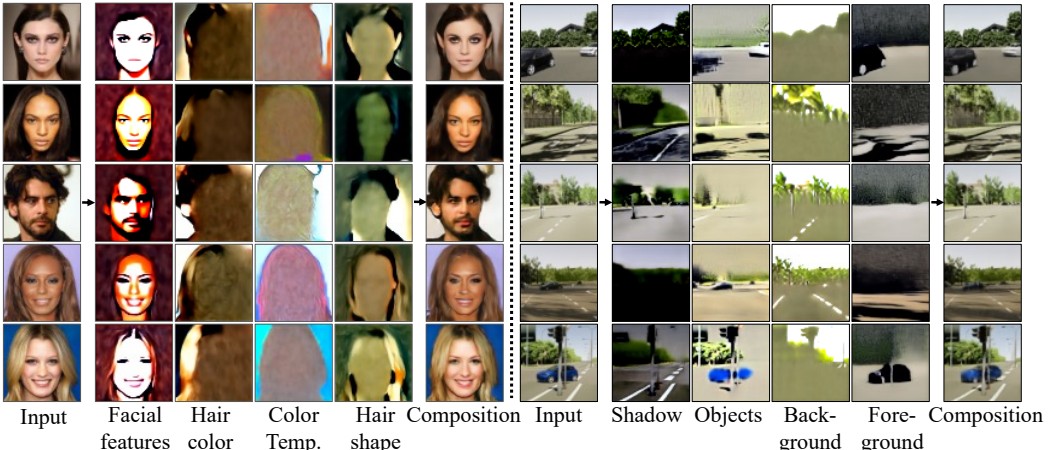

Figure XI: **Global Factor Decomposition.** Global factor decomposition and composition results on CelebA-HQ and Virtual KITTI 2. Note that we name inferred concepts for easier understanding.

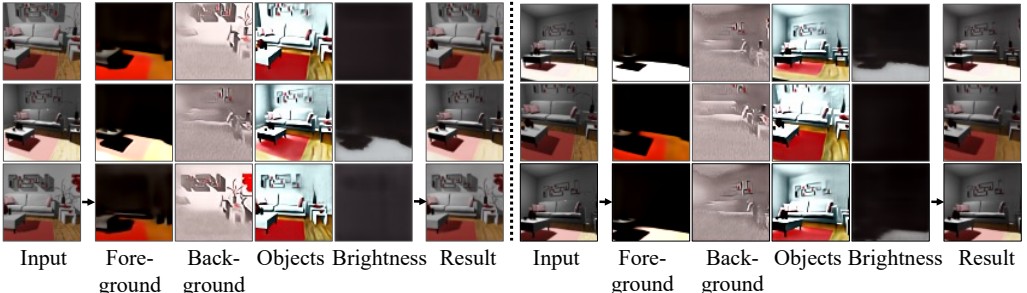

Figure XII: **Global Factor Decomposition.** Global factor decomposition and composition results on Falcor3D. Note that we name inferred concepts for easier understanding.

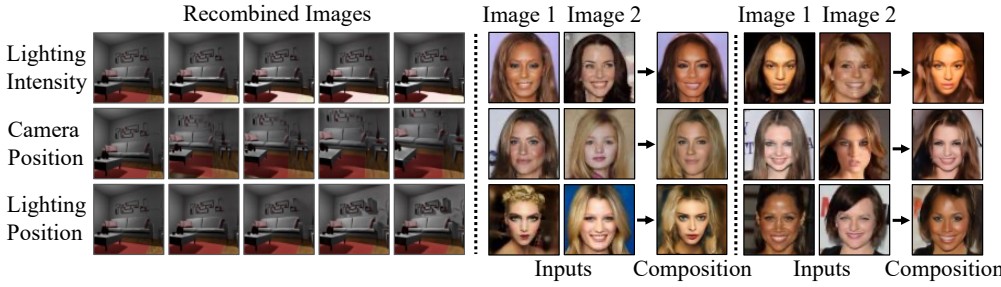

Facial features, Color temp. (Image 1) + Hair color, Hair shape (Image 2)

Figure XIII: **Global Factor Recombination.** Recombination of inferred factors on Falcor3D and CelebA-HQ datasets. In Falcor3D (**Left**), we show image variations by varying inferred factors such as lighting intensity. In CelebA-HQ (**Right**), we recombine factors from two different inputs to generate novel face combinations.

## C    ADDITIONAL EXPERIMENTS

**Impact of the Number of Components K**. We provide qualitative comparisons on the number of components $K$ used to train our models in Figure XVI and Figure XVII.

**Decomposition Comparisons**. We provide qualitative comparisons of decomposed concepts in Figure XVIII and Figure XIX.

## D    MODEL DETAILS

We used the standard U-Net architecture from Ho et al. (2020) as our diffusion model. To condition on each inferred latent $z_k$, we concatenate the time embedding with encoded latent $z_k$, and use that as our input conditioning. In our implementation, we use the same embedding dimension for both time embedding and latent representations. Specifically, we use 256, 256, and 16 as the embedding dimension for both timesteps and latent representations for CelebA-HQ, Virtual KITTI

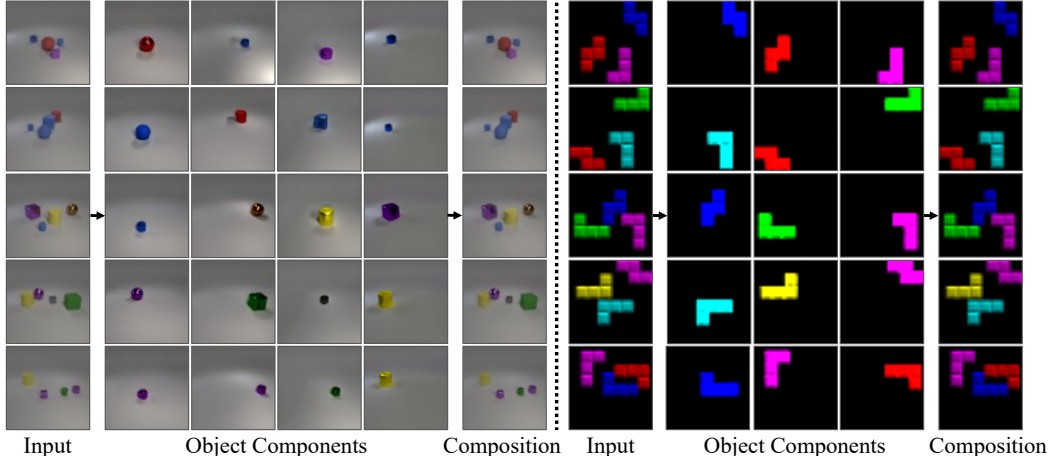

|  Input | Object Components | Composition | Input | Object Components | Composition |

Figure XIV: **Local Factor Decomposition.** Object-level decompositions results on CLEVR (**left**) and Tetris (**right**).

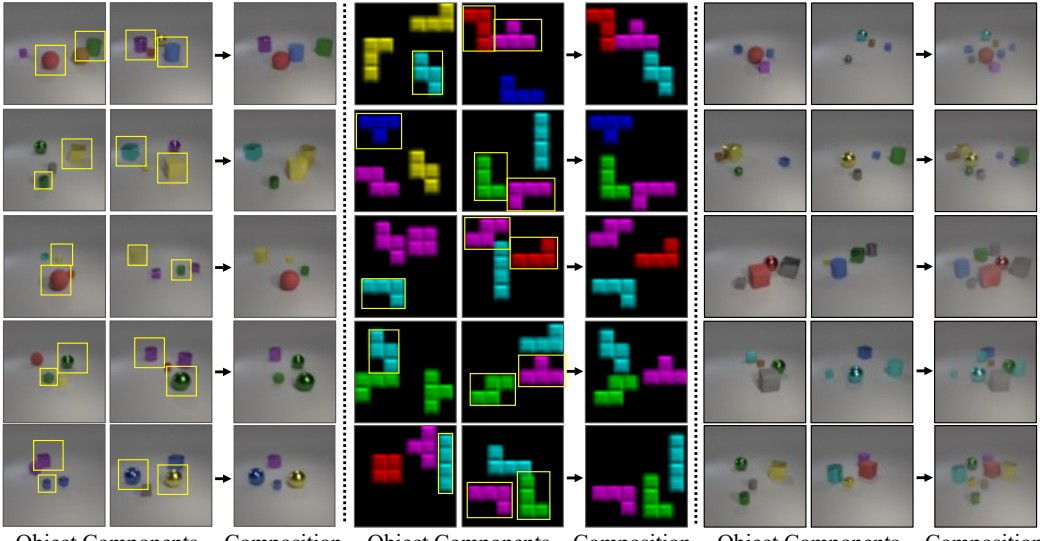

|  Object Components | Composition | Object Components | Composition | Object Components | Composition |

Figure XV: **Local Factor Recombination.** Recombination results using object-level factors from different images.

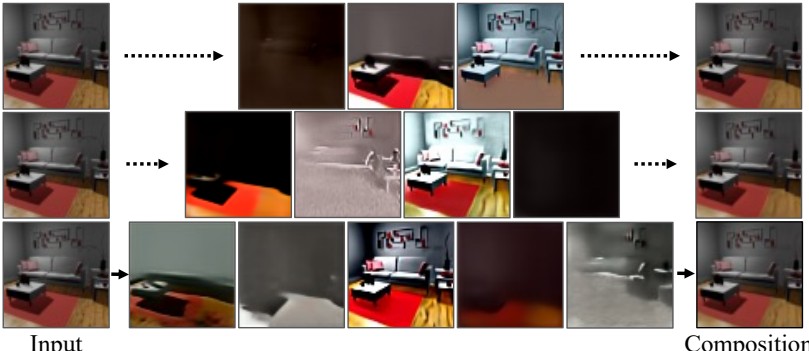

| Input | Composition |

Figure XVI: Decomp Diffusion trained on Falcor3D dataset with varying number of components $K = 3, 4,$ and 5

2, and Falcor3D, respectively. For datasets CLEVR, CLEVR Toy, and Tetris, we use an embedding dimension of 64.

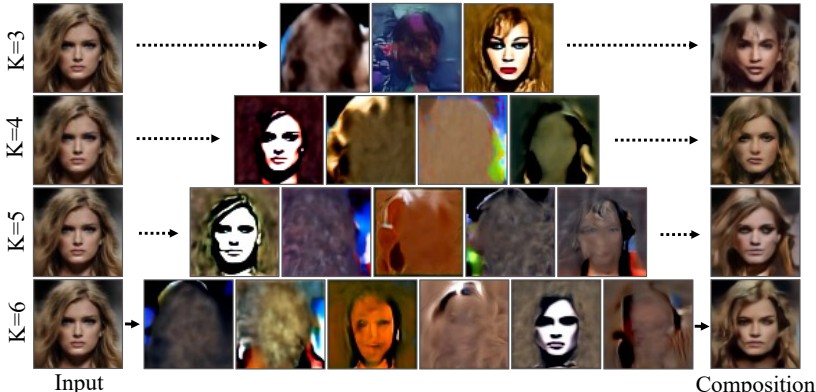

Figure XVII: Decomp Diffusion trained on CelebA-HQ with varying number of components $K = 3, 4, 5,$ and $6$

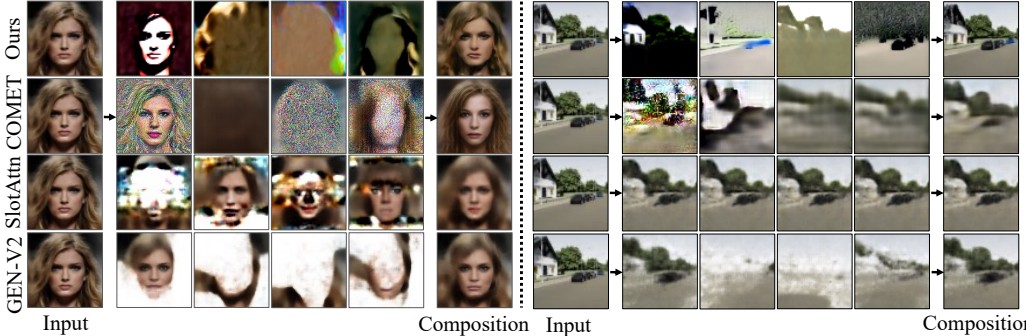

Figure XVIII: **Qualitative comparisons on CelebA-HQ and VKITTI datasets**. Decomposition results on CelebA-HQ (**Left**) and Virtual KITTI 2 (**Right**) on benchmark object representation methods. Compared to our method, COMET generates noisy components and less accurate reconstructions. SlotAttention may produce identical components, and it and GENESIS-V2 cannot disentangle global-level concepts.

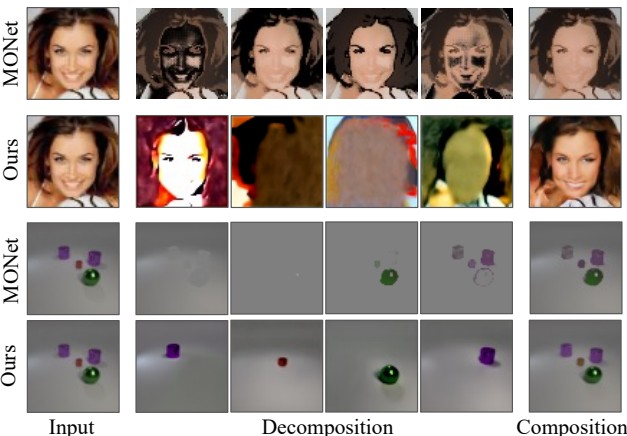

Figure XIX: **Decomposition comparisons on CelebA-HQ and CLEVR datasets**. We further provide qualitative comparison between our method with MONet on decomposition. Our method can decompose images into factors that are more visually diverse and meaningful, while MONet could fail to disentangle factors.

To infer latents, we use a ResNet encoder with hidden dimension of $64$ for Falcor3D, CelebA-HQ, Virtual KITTI 2, and Tetris, and hidden dimension of $128$ for CLEVR and CLEVR Toy. In the encoder, we first process images using 3 ResNet Blocks with kernel size $3 \times 3$. We downsample images between each ResBlock and double the channel dimension. Finally, we flatten the processed residual features and map them to latent vectors of a desired embedding dimension through a linear layer.

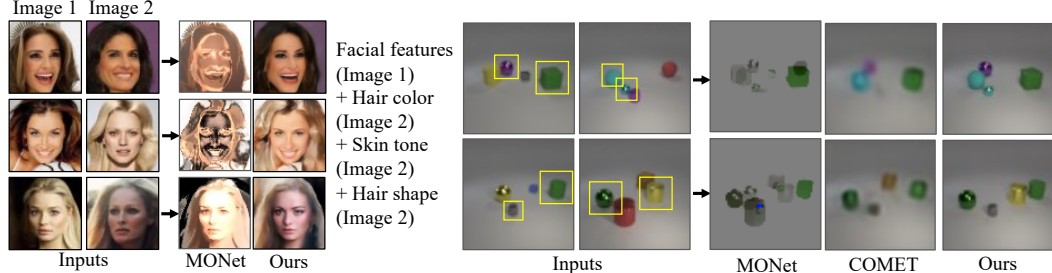

Figure XX: **Recombination comparisons on CelebA-HQ and CLEVR with MONet**. We further compare with MONet on recombination. Our method outperforms MONet by generating correct recombinations results.

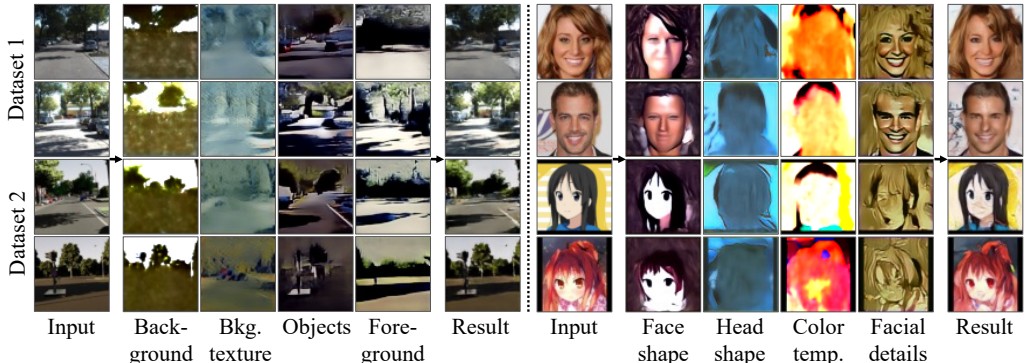

Figure XXI: **Multi-modal Dataset Decomposition.** Multi-model decomposition and composition results on hybrid datasets such as KITTI and Virtual KITTI 2 scenes (**Left**), and CelebA-HQ and Anime faces (**Right**). The top 2 images are of the first dataset, and the bottom 2 images are of the second dataset. Inferred concepts are named for better understanding.

## E  EXPERIMENT DETAILS

In this section, we first provide dataset details in Section E.1. We then describe training details for our baseline methods in Section E.2. Finally, we present training and inference details of our method in Section E.3 and Section E.4.

### E.1  DATASET DETAILS

| CLEVR | CLEVR Toy | CelebA-HQ | Anime | Tetris | Falcor3D | KITTI | Virtual KITTI 2 |
|-------|-----------|-----------|-------|--------|----------|-------|-----------------|
| 10K   | 10K       | 30K       | 30K   | 10K    | 233K     | 8K    | 21K             |

Table IV: **Training dataset sizes.**

Our training approach varies depending on the dataset used. Specifically, we utilize a resolution of $32 \times 32$ for Tetris images, while for other datasets, we use $64 \times 64$ images. The size of our training dataset is presented in Table IV and typically includes all available images unless specified otherwise.

**Anime.** (Branwen et al., 2019) When creating the multi-modal faces dataset, we combined a $30,000$ cropped Anime face images with $30,000$ CelebA-HQ images.

**Tetris.** (Greff et al., 2019) We used a smaller subset of 10K images in training, due to the simplicity of the dataset.

**KITTI.** (Geiger et al., 2012) We used $8,008$ images from a scenario in the the Stereo Evaluation 2012 benchmark in our training.

**Virtual KITTI** 2. (Cabon et al., 2020) We used $21,260$ images from a setting in different camera positions and weather conditions.

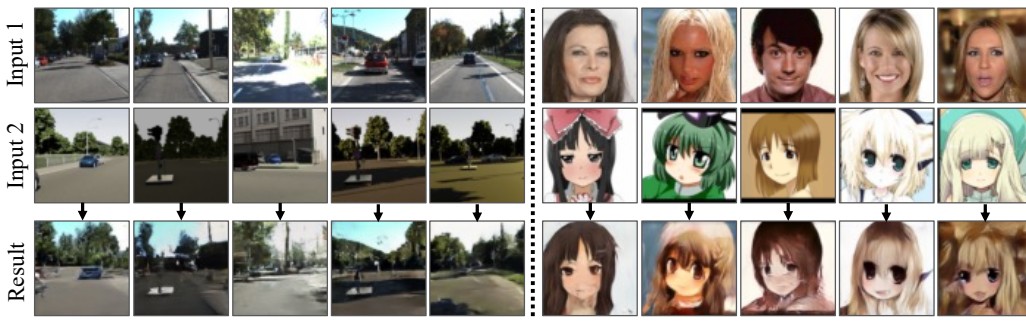

Figure XXII: **Multi-modal Dataset Recombination.** Recombinations of inferred factors from hybrid datasets. We recombine different extracted factors to generate unique compositions of KITTI and Virtual KITTI 2 scenes (**Left**), and compositions of CelebA-HQ and Anime faces (**Right**).

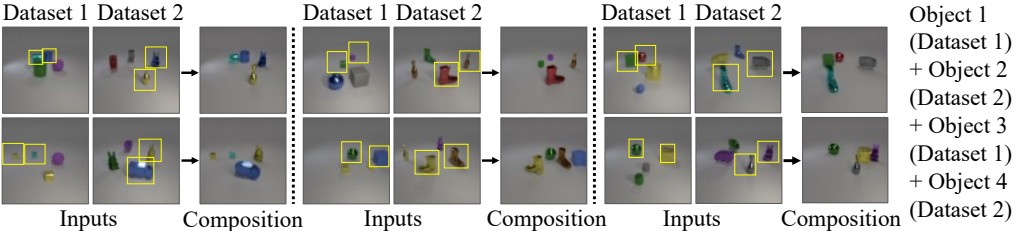

Figure XXIII: **Cross Dataset Recombination.** We further showcase our method's ability to recombine across datasets using 2 different models that train on CLEVR and CLEVR Toy, respectively. We compose inferred factors as shown in the bounding box from two different modalites to generate unseen compositions.

### E.2 BASELINES

**Info-GAN (Chen et al., 2016).** We train Info-GAN using the default training settings from the official codebase at https://github.com/openai/InfoGAN.

**$\beta$-VAE (Higgins et al., 2017).** We utilize an unofficial codebase to train $\beta$-VAE on all datasets til the model converges. We use $\beta = 4$ and $64$ for the dimension of latent $z$. We use the codebase in https://github.com/1Konny/Beta-VAE.

**MONet (Burgess et al., 2019).** We use an existing codebase to train MONet models on all datasets until models converge, where we specifically use $4$ slots, and $64$ for the dimension of latent $z$. We use the codebase in https://github.com/baudm/MONet-pytorch.

**COMET (Du et al., 2021a).** We use the official codebase to train COMET models on various datasets, with a default setting that utilizes $64$ as the dimension for the latent variable $z$. Each model is trained until convergence over a period of $100,000$ iterations. We use the codebase in https://github.com/yilundu/comet.

**SlotAttention (Locatello et al., 2020b).** We use an existing PyTorch implementation to train SlotAttention from https://github.com/evelinehong/slot-attention-pytorch .

**GENESIS-V2 (Engelcke et al., 2021b).** We train GENESIS-V2 using the default training settings from the official codebase at https://github.com/applied-ai-lab/genesis .

### E.3 TRAINING DETAILS

We used standard denoising training to train our denoising networks, with $1000$ diffusion steps and squared cosine beta schedule. In our implementation, the denoising network $\epsilon_\theta$ is trained to directly predict the original image $x_0$, since we show this leads to better performance due to the similarity between our training objective and autoencoder training.

To train our diffusion model that conditions on inferred latents $z_k$, we first utilize the latent encoder to encode input images into features that are further split into a set of latent representations $\{z_1, \ldots, z_K\}$.

| Model | CLEVR | | CLEVR Toy | |
|---|---|---|---|---|
| | FID ↓ | KID ↓ | FID ↓ | KID ↓ |
| COMET | 98.27 | 0.110 | 192.02 | 0.250 |
| Ours | 75.16 | 0.086 | 52.03 | 0.052 |

Table V: **Cross-dataset quantitative metrics.** For evaluating cross-dataset recombination (CLEVR combined with CLEVR Toy), because there is no ground truth for recombined images, we computed FID and KID scores of generated images against the original CLEVR dataset and CLEVR Toy dataset. Our approach achieves better scores for both datasets compared to COMET, which suggests that our generations are more successful in recombining objects from the original datasets.

For each input image, we then train our model conditioned on each decomposed latent factor $z_k$ using standard denoising loss.

Regarding computational cost, our method uses $K$ diffusion models, so the computational cost is $K$ times that of a normal diffusion model. In practice, the method is implemented as $1$ denoising network that conditions on $K$ latents, as opposed to $K$ individual denoising networks. One could significantly reduce computational cost by fixing the earlier part of the network, since latents would only be conditioned on in the second half of the network. This would likely achieve similar results with reduced computation. In principle, we could also parallelize $K$ forward passes to compute $K$ score functions to reduce both training and inference time.

Each model is trained for $24$ hours on an NVIDIA V100 32GB machine or an NVIDIA GeForce RTX 2080 24GB machine. We use a batch size of $32$ when training.

### E.4 INFERENCE DETAILS

When generating images, we use DDIM with 50 steps for faster image generation.

**Decomposition.** To decompose an image $x$, we first pass it into the latent encoder $\text{Enc}_\theta$ to extract out latents $\{z_1, \cdots, z_K\}$. For each latent $z_k$, we generate an image corresponding to that component by running the image generation algorithm on $z_k$.

**Reconstruction.** To reconstruct an image $x$ given latents $\{z_1, \cdots, z_K\}$, in the denoising process, we predict $\epsilon$ by averaging the model outputs conditioned on each individual $z_k$. The final result is a denoised image which incorporates all inferred components, *i.e.*, reconstructs the image.

**Recombination.** To recombine images $x$ and $x'$, we recombine their latents $\{z_1, \cdots, z_K\}$ and $\{z'_1, \cdots, z'_K\}$. We select the desired latents from each image and condition on them in the image generation process, *i.e.*, predict $\epsilon$ in the denoising process by averaging the model outputs conditioned on each individual latent.

To additively combine images $x$ and $x'$ so that the result has all components from both images, *e.g.*, combining two images with $4$ objects to generate an image with 8 objects, we modify the generation procedure. In the denoising process, we assign the predicted $\epsilon$ to be the average over all $2 \times K$ model outputs conditioned on individual latents in $\{z_1, \cdots, z_K\}$ and $\{z'_1, \cdots, z'_K\}$. This results in an image with all components from both input images.

