# OpenReview forum: "Compositional Image Decomposition with Diffusion Models"
_ICLR.cc/2024/Conference — Submitted to ICLR 2024_

### Official Review · Reviewer_t1av · 2023-10-30

**Soundness:** 3 good
**Presentation:** 3 good
**Contribution:** 3 good
**Rating:** 6
**Confidence:** 4

**Summary:**

This paper focuses on the image decomposition task and proposes a new approach, i.e., Decomp Diffusion, to decompose a scene into a set of factors represented as separate diffusion models. The proposed method can decompose scenes into both global and local concepts. These concepts can further be flexibly composed to generate a variety of scenes.

**Strengths:**

The idea of leveraging the connection between Energy-based models and diffusion models for image decomposition is interesting and effective. The compositional concepts from images can be discovered in an unsupervised manner. The experimental results show that the proposed method can discover both global and local concepts, and be used for component compositions across multiple datasets and models.

**Weaknesses:**

1. The quantitative evaluation is not thorough. The current quantitative evaluation only focuses on the global factors, while the quantitative evaluation for the local factors and cross dataset generalization is missing. In contrast, the existing work (COMET) contains quantitative comparisons for the object-level decomposition.
2. As the proposed method contains a set of diffusion models, the computational cost of the proposed method and existing works should be discussed in the paper.
3. For training details in the supplemental, each model is trained on an NVIDIA V100 or an NVIDIA RTX 2080 with the same hours. I was wondering whether the model performance would be different. In addition, is the memory of NVIDIA RTX 2080 24GB or 8GB?

**Questions:**

1. For the ablation study, why use MSE and LPIPS to evaluate the reconstruction quality, rather than the metrics used in Table 1? How about the results of the ablated versions on other datasets used in the paper?
2. How to determine the types of factors that can be inferred from the image? For example, I am not sure whether the second to the fourth columns correspond to shadow, objects, and background respectively.

---

> ### Author Response · Authors · 2023-11-17
> **Reviewer Response**
>
> Thank you for your constructive comments. We have added quantitative metrics for local factor decomposition and cross-dataset generalization as well as for the ablation study. In addition, we answer questions about the computational cost, machine information, and inferring factors in more detail.
>
> **Q1) Quantitative metrics for local factors and cross-dataset generalization**
>
> To quantitatively evaluate our method on local factors, we compare our method with several baselines for local factors (CLEVR) in the table below (we have also updated it in Table 1 in our main paper). Our method achieves the best performance across all three metrics, achieving the best performance on local factor decomposition.
>
> | Model | FID &darr; | KID &darr; | LPIPS &darr;
> |  :----------------  |  :------:  |  :------:  | :------:  |
> | Beta-VAE | 316.64 | 0.383 | 0.651
> | MONet | 60.74 | 0.063 | 0.118
> | COMET | 103.84 | 0.119 | 0.141
> | Slot Attention | 27.07 | 0.026 | 0.031
> | Hessian Penalty | 25.40 | 0.016 | ---
> | GENESIS-V2 | 318.46 | 0.403 | 0.631
> | Ours | **11.49** | **0.011** | **0.012**
>
> For cross-dataset evaluation (CLEVR combined with CLEVR Toy), because there is no ground truth for recombined images, we computed FID and KID scores of generated recombination images against the original CLEVR dataset as well as the original CLEVR Toy dataset. We report in the table below and have added it in the Appendix. Our approach achieves better scores for both original datasets compared to COMET, which suggests that our generations are more successful in recombining objects from the original datasets.
>
> | Model | CLEVR  FID  | CLEVR KID  | CLEVR Toy  FID | CLEVR Toy KID |
> |  :----------------  |  :------:  |  :------:  | :------:  |  :------:  |
> | COMET | 98.27 | 0.110 | 192.02  | 0.250 |
> | Ours | **75.16**  | **0.086** | **52.03**  | **0.052** |
>
>
> **Q2) Computational cost**
> Please see general response Section 4. Our method uses $K$ diffusion models, so the computational cost is $K$ times that of a normal diffusion model. In practice, the method is implemented as $1$ denoising network that conditions on $K$ latents, as opposed to $K$ individual denoising networks. One could significantly reduce computational cost by fixing the earlier part of the network and only condition on latents in the second half of the network. This would likely achieve similar results while reducing computation. In principle, we could also parallelize $K$ forward passes for each factor, thus reducing both training and inference time.  We have added this detail in the Section E.3 of the Appendix. Regarding the computational cost of other approaches, Beta-VAE, MONet, COMET, and Hessian Penalty generally used less training time until convergence, while Slot Attention and GENESIS-V2 use a comparable training time of 24 hours.
>
> **Q3) Machine specs**
> Our models were trained with the same batch size of 32 images, so the machine type should not affect model performance. The memory of a NVIDIA RTX 2080 machine is 24GB.
>
> **Q4) Ablation study metrics**
>
> Thank you for your question. Our method not only decomposes images into a set of factors, but also enables recombination of all inferred factors to reconstruct the input image. As a result, we used MSE and LPIPS to measure both pixel-wise errors and perceptual similarity of the original input image and reconstructed image. In addition to MSE and LPIPS, we have also added FID and KID to the ablation study in the table below and update it in the main paper accordingly (Table 2):
>
> | Dataset | Multiple Components | Predict x_0 | MSE &darr; | LPIPS &darr; | FID &darr; | KID &darr;
> |  :----------------  |  :------:  |  :------:  | :------:  | :------:  | :------:  | :------:  |
> | CelebA-HQ | YES | NO | 105.003 | 0.603 | 155.46 | 0.141
> | CelebA-HQ | NO | YES | 88.551 | 0.192 | 30.10 | 0.022
> | CelebA-HQ | YES | YES | **76.168** | **0.089** | **16.48** | **0.013**
> | CLEVR | YES | NO | 56.179 | 0.3601 | 42.72 | 0.033
> | CLEVR | NO | YES | 26.094 | 0.2236 | 24.27 | 0.023
> | CLEVR | YES | YES | **6.178** | **0.0122** | **11.54** | **0.010**
>
> In the table above, we have also included ablation study for the CLEVR dataset. The non-ablated version of the model, with predicting $x_0$ and using multiple components (i.e., 4 components), achieves the best scores on both CelebA-HQ and CLEVR datasets across all metrics, indicating that the conclusion is consistent across different datasets.
>
>
> **Q5) How are the types of inferred factors determined?**
> Please see general response Section 3. Since our method learns to decompose images in an unsupervised manner, there is no way to know what type of factors would be inferred beforehand. As a result, we provide a large number of example images and name inferred factors through visual inspection in order to help readers understand what each component represents. We have updated the Experiment section to clarify this.

---

### Official Review · Reviewer_Rmft · 2023-10-30

**Soundness:** 2 fair
**Presentation:** 2 fair
**Contribution:** 2 fair
**Rating:** 5
**Confidence:** 4

**Summary:**

This paper addresses compositional image generation through denoising diffusion models. The unsupervised approach decomposes the input image into several primitives, and the model is able to recompose these primitives together. Experiments are conducted on simple object scenes and human faces, and demonstrate superior performance than SOTAs.

**Strengths:**

+ The paper addresses compositional modeling for images using denoising diffusion models. The recomposition quality seems promising.
+ The paper shows that energy functions are additive of primitives.

**Weaknesses:**

+ The method seems to be similar to [1]
+ What is the computational cost? It may takes more space and computational resources with K diffusion models



[1] Du et al, Reduce, Reuse, Recycle: Compositional Generation with Energy-Based Diffusion Models and MCMC, ICML 2023

**Questions:**

+ Is the learned encoder Encθ(x) pre-trained or trained with diffusion model jointly?
+ Suppose it is jointly trained, how does the network learn to decompose the image into shadow image, object image, background image, etc? Is there any specific constraint for learning these different properties?

---

> ### Author Response · Authors · 2023-11-17
> **Reviewer Response**
>
> Thank you for the feedback. We address the comparison to a related work, and answer other questions about the computational cost, encoder, and inferred factors in more detail.
>
> **Q1) The method seems similar to [1]**
> Though both works involve diffusion models representing concepts, the goal of this paper is to discover and decompose images into component diffusion models, while [1] focuses on composing pre-trained diffusion models. To accomplish this decomposition, we propose a new training-time objective to find each component.
>
> **Q2) What is the computational cost?**
> Thank you for your question. Our method computes $K$ conditional scores for a total of $K$ latents, so the computational cost is $K$ times that of a normal diffusion model. In practice, the method is implemented as $1$ denoising network that conditions on $K$ latents, as conditioning on different latents can make arbitrary functions in principle. One could significantly reduce computational cost by fixing the earlier part of the network since latents are only conditioned in the second half of the network. In addition, in practice, we parallelize $K$ forward passes for each factor, thus keeping training and inference time constant.
>
> We have added this detail in the Section E.3 of the Appendix.
>
>
> **Q3) How is the encoder learned?**
> Our method jointly learns the latent encoder and the denoising network jointly. We have also updated Section 3.2 in the paper to clarify this choice.
>
> **Q4) How does the model learn to decompose images into different factors?**
> We have described how the model learns to disentangle components in general response Section 2, and explained how we name our inferred factors in general response Section 3. To further clarify, the model learns to extract meaningful latents primarily through the information bottleneck principle. The encoder, which is jointly trained, compresses the inputs $x$ into a latent representation $z$ consisting of $K$ low-dimensional sub-latents $z_k$. Since these sub-latents are very low-dimensional, to most effectively denoise images, each of the different functions $\epsilon_\theta(x_i^t, t, z_k)$, focuses on different aspects of an image (as if any two functions focus on the same aspect, then the information capacity between the two latents is not being effectively used). This encourages disentanglement in the learned functions across $K$ sub-latent representations, where each sub-latent captures specific, independent aspects of the input images $x$. An interesting direction of future work is to add other constraints to enforce orthogonality between different sub-latents, though we had limited success when trying this. We have added this information in Section 3.2.
>
>
> Since our method learns to decompose images in an unsupervised manner, there is no way of knowing what type of factors would be inferred beforehand. As a result, we provide a large set of example images and name inferred factors through visual inspection in order to help readers understand what each component represents. We have updated the Experiment section to clarify this.

---

> > ### Comment · Reviewer_Rmft · 2023-11-18
> > **Response**
> >
> > Thank you for your reply. It is good to see that the encoder and decoder are trained jointly to decompose an image into a fixed number of elements. I am curious to see what would it be to use GANs to unsupervised decompose an image?

---

> > > ### Author Response · Authors · 2023-11-18
> > > **Response**
> > >
> > > Thank you for your response. We don’t think there is an easy way use a GAN to discover a set of composable components across datasets, as there is no easy way to compose GANs together. However, you can use a GAN to discover a latent space for images. We actually compare to one such GAN (Hessian Penalty [1]) in the main paper and find that our approach outperforms this baseline.
> > >
> > > [1] Peebles et al. The Hessian Penalty: A Weak Prior for Unsupervised Disentanglement

---

> > > > ### Comment · Reviewer_Rmft · 2023-11-18
> > > > **Response**
> > > >
> > > > Yes, this is a good point. GANs may have mode collapse problem. Now let's say VAE. VAE can be additive to composition. I assume VAE can also decompose an image into different primitives, right?

---

> > > > > ### Author Response · Authors · 2023-11-19
> > > > > **Response**
> > > > >
> > > > > Thank you for your quick response — yes you can  also use a VAE also to decompose an image into a set of components. We ran several comparisons with different VAEs, both beta-VAE [1] and MONET [2]. We find that our approach also significantly outperforms these two baselines. MONET implements compositions by linearily adding components together — in contrast our approach can more flexibly combine components together through optimization.
> > > > >
> > > > > [1] Higgins et al. beta-VAE: Learning Basic Visual Concepts with a Constrained Variational Framework
> > > > > [2] Burgess et al. MONet: Unsupervised Scene Decomposition and Representation

---

> > > > > > ### Comment · Reviewer_Rmft · 2023-11-19
> > > > > > **Response**
> > > > > >
> > > > > > This is good. However, I only find quantitative comparison on reconstruction quality in Table 1. I do not see any visual comparisons of disentanglement of primitives, which I believe is quite important to validate your method.

---

> ### Author Response · Authors · 2023-11-20
> **Response**
>
> Hi, yes in Figure XVIII of the original paper you can see a visual comparison of disentanglement of primitives with baselines. We've also just added a revision to the paper to provide additional visual comparisons of disentanglement with MONet in Figure XIX ([screenshot here](https://ibb.co/Z6Qssyt)) and recombination in Figure XX ([screenshot here](https://ibb.co/pQH0JF6)).

---

> > ### Author Response · Authors · 2023-11-22
> > **Author Followup**
> >
> > Hi Reviewer Rmft,
> >
> > Thank you for your time reviewing. As the discussion period ends today, we wanted to check if you had any other concerns or clarifications.
> >
> > Thanks,
> >
> > Paper Authors

---

### Official Review · Reviewer_W4iB · 2023-11-01

**Soundness:** 3 good
**Presentation:** 3 good
**Contribution:** 3 good
**Rating:** 8
**Confidence:** 4

**Summary:**

The paper addresses unsupervised image decomposition/re-composition with diffusion models. The authors show equivalence between previous decomposition work COMET, based on energy minimization and gradient descent optimization framework, and the recent diffusion models (DDPM) denoising steps iteration.  They consequently 'substitute' the EM model with a diffusion model conditionned on a set of latent variables z_k. The z_k's are inferred by an Encoder, and are associated to the different factors of the decomposition.
Experimental results are illustrated on several classical benchmarks (CelebA, Virtual Kitti, Falcor3D, also synthetic data such as CLEVR and Tetris), compared qualitatively and quantitatively to related work.

**Strengths:**

Unsupervised image intrinsic decomposition/re-composition is very challenging and one of the most fundamental open issues in computer vision. Using diffusion models for this purpose seems a natural choice (given the success of DM in natural image generation, and in learning semantic image properties). The authors give a rigorous justification of their choices from a mathematical point of view.  The paper's idea is well argued. The illustrated results show the strong potential of the approach.  I enjoyed reading the article.

**Weaknesses:**

Qualitative results are promising but still leave room for improvement. Reconstructed images appear blurry, and at low resolution. But at this stage this is not a major issue and that might be improved by further work.

**Questions:**

1) I did not find in the paper an explanation about the encoder z = Enc_\theta(x).  How Enc_() is learnt? What ensures that the decomposition is at the local (ie objects, things) or global (ie, illumination, stuffs) level?   What ensures the disentanglement of the decomposition ? (no additional constraints are enforced during the learning stage).  Those aspects might have been discussed in the original paper COMET (I did not read it), however, it is worth to discuss them again in the current paper since it is key for the understanding and  analysis of the success/failures of the proposed approach.

2) Some details in the approach that are not clear to me.
2.1 The authors argue that they 'learn a set of different denoising functions to recover an image x_i' (page 4). However, the denoising function \epsilon_\theta is not, in eq.9, parameterized by k.  The only dependance to k is in the input latent variable z_k. It would imply that there is a single denoising function, but with different input argument (in particular the z_k). Please clarify.
2.2 The encoder Enc_\theta() and the denoising function \epsilon_\theta, are both parameterized by \theta. This is probably a typo, the two networks being parameterized by two sets of independent weights, \theta_1 and \theta_2. Please correct as needed in the paper.

**Details Of Ethics Concerns:**

No ethical issues beyond existing public generative models.

---

> ### Author Response · Authors · 2023-11-17
> **Reviewer Response**
>
> Thank you for your detailed comments and positive evaluation. We have added a description of the encoder training and explanation of how the approach learns to disentangle latents in the text.
>
> **Q1) How is the encoder learned?**
> Thank you for your question. As addressed in the general response 1, our encoder and denoising network are learned jointly using the denoising loss. The main reason behind this design choice is that we can naturally enable the model to learn different types of latent representations from various datasets.
>
> We have also clarified this in the method section accordingly.
>
> **Q2) What ensures the type of decomposition?**
> Since our method is an unsupervised approach to decompose images into a set of latents, we aren’t able to know the types of inferred concepts those learned latents would be ahead of time. Based off visual inspection, we provided a 'psuedolabel' for each component based off the visualization of the component. To more clearly display each component, we provide a set of visualizations in our experiments and appendix to help readers understand inferred concepts.
>
> We have also updated this in the Experiment section to avoid any further confusion.
>
> **Q3) How is disentanglement ensured?**
> Please see our general response 2. Disentanglement is primarily ensured by information bottleneck. The encoder encodes the information in $x$ into a compact low-dimensional latent representation $z$ consisting of $K$ sub-latents $z_k$. To most effectively denoise an image given these low-dimensional latents, each function $\epsilon_\theta(x_i^t, t, z_k)$ learns to focus on distinct information in an image. This enforces diversity and disentanglement in the learned functions across $K$ sub-latent representations. While we explored additional constraints to enforce independence in components, we found that it gave similar performance to just using a low dimensional latent.
>
> We have added this information in Section 3.2.
>
>
> **Q3) Why is one single denoising network used instead of $K$**
> Thank you for your question. In principle, a model can learn arbitrary functions by conditioning on different latent representations. Thus one single denoising network that conditions on $K$ different latents should achieve similar performance as $K$ denoising networks. In addition, our method is more memory-efficient as it only requires one network instead of $K$.
>
> **Q4) Typos**
> Thank you for pointing that out. We have updated the encoder parametrization notation to be $Enc_\phi$ instead of $Enc_\theta$. We will make sure that our notations are consistent in our next version of the paper.

---

> > ### Comment · Reviewer_W4iB · 2023-11-23
> >
> > Thank you to the authors for the clarifications. They mainly address my concerns.
> > The approach developed in the paper is theoretically well founded and the main concept clearly justified/discussed. Despite its limitations,  I do believe that the paper paves the way for future work.

---

### Author Response · Authors · 2023-11-17
**General Response**

We thank reviewers for their detailed comments and feedback. We’re glad that reviewers noted that our work is interesting and effective (reviewer t1av), and illustrated results are promising (reviewer Rmft) and show a great potential of the approach (reviewer W4ib).

Reviewers had concerns with some of the method details, including how the encoder is learned in our approach (Reviewer W4iB, Rmft), what ensures the disentanglement of decomposition (Reviewer W4iB), how to determine the types of inferred factors (Reviewer W4iB, Rmft, t1av) and the potential efficiency of the approach, such as the computational costs of training and inference (Reviewer Rmft, t1av) which we discuss below. We address these in  individual reviewer’s questions. For convenience, we highlight updated text in the paper in blue.

## 1. How the encoder is learned
Our encoder and denoising network are learned jointly in our experiments. By doing so, we naturally enable the model to learn different types of latent representations, i.e., global or local representations, from various input images. We have also updated it in the method section accordingly.

## 2. What ensures the disentanglement of decomposition
The encoder compresses the inputs $x$ into a set of low-dimensional latent representations $z$ consisting of $K$ sub-latents $z_k$. By information bottleneck, as the latent dimensionality is very small, these individual latent are encouraged to discover orthogonal and important information from the inputs to best enable effective denoising across the different denoising functions $\epsilon_\theta(x_i^t, t, z_k)$. This constraint encourages diversity and disentanglement in the learned functions across the $K$ sub-latent representations, so that the functions can best jointly denoise the input images $x$ (repetition in factors between functions would lead to wasted latent capacity to reconstructing images).  While we tried adding additional losses to more explicitly enforce independence between latents, we found that this latent dimensionality constraint was sufficient. We have added this information in Section 3.2.

## 3. How to determine the types of inferred factors
Since our method is an unsupervised method to decompose images into a set of factors, there is no way to know what type of factors would be inferred beforehand. As a result, we provide an extensive set of visualizations and name individual inferred factors after manual inspection in order to help readers understand what each component represents. We have updated the Experiment section to clarify this concern.

## 4. Computational cost
Regarding questions about the computational cost, the method uses $K$ diffusion models, so the computational cost is $K$ times that of a normal diffusion model. In practice, the method is implemented as $1$ denoising network that conditions on $K$ latents, as opposed to $K$ individual denoising networks. One could significantly reduce computational cost by fixing the earlier part of the network, since latents will only be conditioned on in the second half of the network. This would likely achieve similar results with reduced computation. In practice, we parallelize $K$ forward passes for each factor, thus reducing both training and inference time to be the same as 1 network.
We have added this detail in the Section E.3 of the Appendix.

## 5. Additional quantitative evaluation
As requested by reviewer t1av, we have also provided quantitative evaluations on local factor decomposition and cross dataset recombination, as well as additional results on diffusion parametrization ablation on the CLEVR dataset. Please see more under the corresponding reviewer’s response section.

---

### Meta-Review · Area_Chair_86Ex · 2023-12-17

**Metareview:**

The reviewers did not reach consensus for the paper. R# W4iB supported the paper with a good sentiment, e.g. unsupervised image intrinsic decomposition is challenging and fundamental (it’s a very subjective statement), and the authors gave a rigorous justification of their model choice and the illustrated results showed strong potential of the approach (again, it’s a vague statement that didn’t articulate why and how). Although R# W4iB gave a rating of 8, the rating is not backed up by clear evidence.

R# t1av is slightly positive over the paper. While acknowledging positive experimental results (but didn’t explain clearly), the reviewer raised concerns of lack of quantitative evaluation and discussion of computational cost. In the rebuttal, the authors provided detailed evidence of how the proposed method outperformed previous methods for local factors and cross-dataset generalization.

R# Rmft is slightly negative over the paper. While the initial concerns were addressed through multiple iterations of discussion, in the final message to the AC, the reviewer committed “Admittedly decomposition performance is improved, I still doubt the unsatisfactory decomposition visual quality, reasonableness and unclear explainability of decomposition primitives.”

The AC’s biggest concern of the paper (and also the direction of this sort of unsupervised image factorization) is that the semantic explanation of these components seems to be very weak and the visual quality is far behind SOTA diffusion models. For all the face examples, the decomposition seems to sometimes lose nose/mouth (Figure 3), but somehow the composition “magically” got them back. The semantics were not verified by diverse examples, e.g. using different hair colors to show that the component is indeed “hair color.” The naming seems to be too arbitrary to be convincing. For “local factor decomposition”, only toy examples with uniform background are used. It’s unclear how the proposed examples can deal with many objects and complex backgrounds. Maybe the “potential” only stays as an illusion. Therefore, the AC recommends reject.

**Justification For Why Not Higher Score:**

The results are just poor. Maybe some reviewers were using the lens of traditional image factorization to judge this paper, but we cannot ignore the amazing image quality from the latest diffusion models, which far surpassed the visual quality of this paper. Also, the naming of the components seems so arbitrary, and it's unclear how useful these components are as the composition quality is very low.

**Justification For Why Not Lower Score:**

N/A

---

### Decision · Program_Chairs · 2024-01-16

Reject